

# Regulation of dye-decolorizing peroxidase gene expression in *Pleurotus ostreatus* grown on glycerol as the carbon source

Jorge Cuamatzi-Flores[1], Soley Nava-Galicia[1], Edgardo Ulises Esquivel-Naranjo[2], Agustin Lopez Munguia[3], Analilia Arroyo-Becerra[1], Miguel Angel Villalobos-López[1] and Martha Bibbins-Martínez[1]

[1] Centro de Investigación en Biotecnología Aplicada-Instituto Politécnico Nacional, Tlaxcala, México
[2] Facultad de Ciencias Naturales, Universidad Autónoma de Querétaro, Querétaro, México
[3] Instituto de Biotecnología, Universidad Autónoma de México, Cuernavaca, Morelos, México

Corresponding author
Martha Bibbins-Martínez,
mbibbinsm@ipn.mx

## ABSTRACT

Dye-decolorizing peroxidases (DyPs) (E.C. 1.11.1.19) are heme peroxidases that catalyze oxygen transfer reactions similarly to oxygenases. DyPs utilize hydrogen peroxide ($H_2O_2$) both as an electron acceptor co-substrate and as an electron donor when oxidized to their respective radicals. The production of both DyPs and lignin-modifying enzymes (LMEs) is regulated by the carbon source, although less readily metabolizable carbon sources do improve LME production. The present study analyzed the effect of glycerol on *Pleurotus ostreatus* growth, total DyP activity, and the expression of three *Pleos-dyp* genes (*Pleos-dyp1*, *Pleos-dyp2* and *Pleos-dyp4*), *via* real-time RT-qPCR, monitoring the time course of *P. ostreatus* cultures supplemented with either glycerol or glucose and Acetyl Yellow G (AYG) dye. The results obtained indicate that glycerol negatively affects *P. ostreatus* growth, giving a biomass production of 5.31 and 5.62 g/L with respective growth rates (micra; m) of 0.027 and 0.023 h$^{-1}$ for fermentations in the absence and presence of AYG dye. In contrast, respective biomass production levels of 7.09 and 7.20 g/L and growth rates ($\mu$) of 0.033 and 0.047 h$^{-1}$ were observed in equivalent control fermentations conducted with glucose in the absence and presence of AYG dye. Higher DyP activity levels, 4,043 and 4,902 IU/L, were obtained for fermentations conducted on glycerol, equivalent to 2.6-fold and 3.16-fold higher than the activity observed when glucose is used as the carbon source. The differential regulation of the DyP-encoding genes in *P. ostreatu*s were explored, evaluating the carbon source, the growth phase, and the influence of the dye. The global analysis of the expression patterns throughout the fermentation showed the up- and down- regulation of the three *Pleos-dyp* genes evaluated. The highest induction observed for the control media was that found for the *Pleos-dyp1* gene, which is equivalent to an 11.1-fold increase in relative expression (log$_2$) during the stationary phase of the culture (360 h), and for the glucose/AYG media was *Pleos-dyp-4* with 8.28-fold increase after 168 h. In addition, glycerol preferentially induced the *Pleos-dyp1* and *Pleos-dyp2* genes, leading to respective 11.61 and 4.28-fold increases after 144 h. After 360 and 504 h of culture, 12.86 and 4.02-fold increases were observed in the induction levels presented by *Pleos-dyp1* and *Pleos-dyp2*, respectively, in the presence of AYG. When transcription levels were referred to those found in the control media, adding AYG led to

up-regulation of the three *dyp* genes throughout the fermentation. Contrary to the fermentation with glycerol, where up- and down-regulation was observed.

The present study is the first report describing the effect of a less-metabolizable carbon source, such as glycerol, on the differential expression of DyP-encoding genes and their corresponding activity.

# INTRODUCTION

The ligninolytic enzymes of white rot fungi are mainly produced during the activation of the secondary metabolism, which occurs under limiting conditions, such as the concentration and presence of bioavailable carbon and/or nitrogen sources (*Alfaro et al., 2020*; *Aro, Pakula & Penttilä, 2005*; *Mikiashvili et al., 2006*; *Stajić et al., 2006*). It has been reported that the ligninolytic isoenzymes encoded by members of a gene family often exhibit variations in their differential expression, catalytic properties, regulation mechanisms, and cellular location (*Fernández-Fueyo et al., 2014*; *Garrido-Bazán et al., 2016*). Furthermore, analysis conducted on the promoters of ligninolytic enzymes encoding genes in the *P. ostreatus* genome has revealed the presence of different putative responsive elements (*Janusz et al., 2013*; *Knop, Yarden & Hadar, 2015*; *Piscitelli et al., 2011*). These elements include carbon catabolite repressor binding elements (CRE), nitrogen response ($Nit_2$), xenobiotic-response elements (XRE), metal-response elements (MRE), and heat-shock elements (HSE), among others, which may be involved in the regulation of gene expression in response to environmental conditions, such as carbon and nitrogen sources or xenobiotics, temperature, and pH (*Jiao et al., 2018*; *Todd et al., 2014*). Carbon catabolite repression (CCR), in combination with different signaling pathways plays a crucial role in the utilization of different carbon sources by *P. ostreatus* and other Basidiomycota fungi (*Daly et al., 2019*; *Toyokawa et al., 2016*; *Yoav et al., 2018*). Furthermore, it has been shown that Cre1, the main transcriptional regulator in the CCR pathway, is regulated by the cAMP-dependent protein kinase A(PKA) (*de Assis et al., 2020*; *Pareek et al., 2022*), while both Cre1 and PKA may be involved in the induction of genes that encode lignin-modifying enzymes in *P. ostreatus* (*Toyokawa et al., 2016*). This process can be studied utilizing carbon sources alternative to glucose, such as glycerol, and evaluating their effect on enzyme activity and/or gene expression.

As part of its wood degradation system *P. ostreatus* produces dye-decolorizing peroxidases (DyPs (EC 1.11.1.19)). These heme peroxidases, can degrade several anthraquinone dyes, and utilize the heme group as a redox cofactor to catalyze the hydrogen peroxide-mediated oxidation of a wide range of molecules, including dyes, namely aromatic and lignin model compounds, some of which are poorly metabolized by other heme peroxidases (*Catucci et al., 2020*; *Singh & Eltis, 2015*; *Xu et al., 2021*). Within the *P. ostreatus* genome, four DyP genes coding for dye-decolorizing peroxidase activity have been identified: *Pleos-dyp1*; *Pleos-dyp2*; *Pleos-dyp3*; and *Pleos-dyp4*
(*Ruiz-Dueñas et al., 2011*). To date, limited reports are available on the factors that regulate DyP production. In a previous study, our research group explored the effect of dyes on the differential expression of *P. ostreatus* DyP-encoding genes and DyP activity, showing that dyes had an induction effect on DyP activity (*Cuamatzi-Flores et al., 2019*).

Additionally, an extracellular proteome analysis conducted during *P. ostreatus* growth on lignocellulosic material revealed the exclusive synthesis of *Pleos*-DyP4 with several versatile peroxidase (VP) and manganese peroxidase (MnP) enzymes (*Fernández-Fueyo et al., 2015*). Glycerol can be a carbon and energy source for several basidiomycetes, including *P. ostreatus*. Furthermore, the activity of some LMEs increases when glycerol or other less-metabolizable carbon sources are used instead of glucose, which could imply that glycerol mediates the carbon catabolite de-repression of LMEs. Given the physiological relevance of DyP enzymes for several groups of organisms and their potential biotechnological applications, the present research aims to investigate the impact of glycerol as a carbon source on the production and differential regulation of DyPs in *P. ostreatus*.

## MATERIALS AND METHODS

### Microorganism

*P. ostreatus* obtained from the American Type Culture Collection (ATCC 32783) (Manassas, VA, USA) was used in the present research. The white rot fungus strain was grown and maintained on potato dextrose agar (PDA).

### Dye decolorization on agar plate

Petri dishes containing agar 15 g/L, glucose (Glu), or glycerol (Gly) as the carbon source and 500 ppm of either Acetyl Yellow G (AYG) (dye content 95%) (Sigma-Aldrich 250309; Sigma-Aldrich, St. Louis, MO, USA), RBBR (Remazol brilliant blue R dye, dye content 50%, Sigma-Aldrich R8001; Sigma-Aldrich, St. Louis, MO, USA), or AB129 (Acid blue 129, dye content 25%, Sigma-Aldrich 306495; Sigma-Aldrich, St. Louis, MO, USA) were inoculated with 0.4 cm$^2$ mycelia plugs taken from the periphery of a *P. ostreatus* colony growing on PDA at 25 °C and then incubated for 7 days. The inoculum was placed, mycelium facing down, on the center of the plate. The plates were then incubated at 25 °C for 8 days. The fungal colony growth and the effect on the dyes were documented using daily photographs throughout the incubation period. The mycelium growth rates ($kr$) were calculated by fitting the linear growth function $y = kr\,x + c$ (where $y$ is the distance and $x$ is the time), expressed in millimeters per day (mm/d) (*Zervakis et al., 2001*), and monitored by carbon source and dye type, with any changes then statistically compared using the Kruskal-Wallis test with the R statistical software, version 4.3.0 (*R Development Core Team, 2023*). The experiments were conducted on two independent replicates.

### Submerged culture conditions and characterization of growth kinetics

The composition of the medium and the conditions for the submerged cultures were established in line with those described by *Cuamatzi-Flores et al. (2019)*. The Acetyl Yellow G dye was selected based on the positive outcome reported for the use of this dye on total

peroxidase activity and expression profiles (*Cuamatzi-Flores et al., 2019*; *Garrido-Bazán et al., 2016*). The present study conducted four types of *P. ostreatus* cultures, using either glucose or glycerol as the carbon source and then adding 500 ppm of Acetyl Yellow G (GAYG and GlyAYG) (dye content 95%) (Sigma-Aldrich 250309; Sigma-Aldrich, St. Louis, MO, USA). Each flask out of three per fermentation type was inoculated with three mycelial plugs (4 mm in diameter) taken using a steel punch from the periphery of *P. ostreatus* colonies grown for 7 d at 25 °C in Petri dishes containing PDA. The cultures were incubated at 25 °C for 23 days on a rotary shaker (SEV-PRENDO 650M) set for constant shaking at 120 rpm. Three flasks were taken as samples every 24 h from 120 (5 d) to 552 h (23 d) of fermentation. The supernatant was obtained by filtering the cultures, using Whatman No. 4 filter paper, and then stored at −20 °C. The glucose level was determined *via* the DNS Method (*Miller, 1959*), while glycerol consumption was assessed as described by *Kuhn et al. (2015)*. The carbon consumption rate was determined by applying the equation $q_S = Y_{X/S} \cdot m$, where $Y_{X/S}$ denotes the yield of gram of biomass per gram of substrate and $\mu$ represents the growth-specific rate (*Dietzsch, Spadiut & Herwig, 2011*). The yield coefficient $Y_{X/S}$ was computed by assessing the ratio of the maximum attained biomass to the corresponding maximum quantity of the carbon source used. Subsequently, the carbon source consumption rate $q_S$ was obtained by multiplying $Y_{X/S}$ by $\mu$ under the prevailing experimental conditions.

Dye-decolorizing peroxidase activity was measured by monitoring the degradation of ABTS (*Salvachúa et al., 2013*), while the percentage of dye decolorization was determined at fixed time intervals, as proposed by *Upadhyay & Przystas (2023)*. The mycelium was rinsed with 0.9% NaCl and stored at −70 °C until subjected to total RNA extraction or dry weight measurement (X, g/L). The specific growth rate ($\mu$) was obtained for each replicate from the logistic equation ($X = X_{max}/(1 + (X_{max} − X_0/X_0) \cdot e^{−m \cdot t})$) using 100 permutations in the R software, version 4.3.0 (*R Development Core Team, 2023*).

The decolorization of the AYG dye was monitored spectrophotometrically at $\lambda_{max}$ (390 nm). All experiments were performed in triplicate. The growth curves were established using the dry biomass measurements obtained from each fermentation.

## RNA extraction and RT-qPCR

The total RNA was isolated from frozen mycelia harvested at different time points during the fermentation, using NTES (100 mM NaCl, 10 mM Tris-HCl, pH 7.5, 1 mM EDTA, and 1% SDS) extraction buffer and a protocol modified from that proposed by *Holding et al. (2007)*. The mycelium was ground in a mortar with liquid nitrogen, with approximately 100 mg of the mycelium then placed in 1.5 mL Eppendorf RNase-free tubes, to which 500 µl of NTES and 500 µl of phenol/chloroform (1:1) were added and stirred until homogenization was complete. The aqueous phase was separated *via* centrifugation at 10,000 rpm and 4 °C for 10 min and then re-extracted using phenol/chloroform. The nucleic acids were precipitated using two volumes of ethanol and 1/10 volume of 2 M sodium acetate, pH 5.3, incubated at −20 °C for 2 h and then resuspended in 250 µl of RNase-free deionized water. The RNA was precipitated using one volume of 4 M LiCl at

**Table 1 Primers used in this study.**

| Gene | Transcript ID[a] | Direction[b] | Sequence (5′ to 3′) | Product size (bp) | Efficiency[c] |
|------|------------------|--------------|---------------------|-------------------|---------------|
| *Pleos-dyp1* | 62271 | Fw | CGCTTGAGTTGATCCAGAAA | 104 | 2.21 |
| | | Rv | TATTTCCTTCGGCTTCCTCA | | |
| *Pleos-dyp2* | 1092668 | Fw | TACATTCTTGCCGCTGGAT | 117 | 1.87 |
| | | Rv | GCGAGAACCTGCTTGAACTT | | |
| *Pleos-dyp4* | 1069077 | Fw | ATGAACACTTCGGCTTCCTC | 64 | 2.03 |
| | | Rv | GGCAAGTACCGCAGATAAG | | |
| *Reference gene (pep)* | 1092697 | Fw | CGGAGGACATTCTTGTTCAC | 142 | 1.89 |
| | | Rv | AGATCGGTAACCCACACGAG | | |

**Notes:**
qPCR primers (type—forward or reverse—and sequences), amplification length and efficiency for the *P. ostreatus* dye peroxidase genes and the selected reference gene, peptidase (*pep*).
[a] Transcript ID and gene nomenclature refer to the annotation of *P. ostreatus* PC15 genome version 2.0 (http://genome.jgi-psf.org/PleosPC15_2/PleosPC15_2.home.html).
[b] Fw, Forward; Rv, reverse.
[c] Efficiency for primers used in qPCR.

−20 °C overnight and then resuspended in the appropriate volume of RNase-free water. The concentration was quantified spectrophotometrically, while the purity was determined using the absorbance ratio at OD 260/280 nm. The RNA was treated with RNAse-free DNase I (Invitrogen, Waltham, MA, USA). The final RNA concentration was set to 300 ng/μl, after which 3 μg of total RNA was reverse-transcribed into cDNA in a volume of 20 μl, using M-MuLV reverse transcriptase (Fermentas, Burlington, Canada), following the manufacturer's protocol.

The RT-qPCR reactions were performed in a StepOne Plus thermal cycler (Applied Biosystems, Waltham, MA, USA), using Maxima SYBR Green/ROX qPCR Master Mix (Thermo Fisher, Waltham, MA, USA) to detect amplification. Specific primers were designed to amplify the transcripts of the three *Pleos-dyp* genes identified in the genome (Table 1). The reaction mixture, amplification program, melting curve, and selection of the reference genes applied adhered to that previously described by *Garrido-Bazán et al. (2016)*. According to their expression stability under the culture conditions of interest and the reference index, consisting of the geometric mean of the best-performing housekeeping genes, the peptidase (*pep*) gene was used for RT-qPCR data normalization. The RT-qPCR reactions were carried out in triplicates with a template-free negative control performed in parallel.

## Promoter sequence analysis

The analysis of regulatory cis-elements in the promoter regions of *Pleos-dyp* genes (Table S1) involved analyzing 2,000-bp upstream genomic DNA sequences of the start codon of each *Pleos-dyp* gene using MEME (Multiple Expectation Maximization for Motif Elicitation, http://meme-suite.org/) based on default parameters. The cis-elements identified were then annotated using SMART (http://smart.embl-heidelberg.de/).

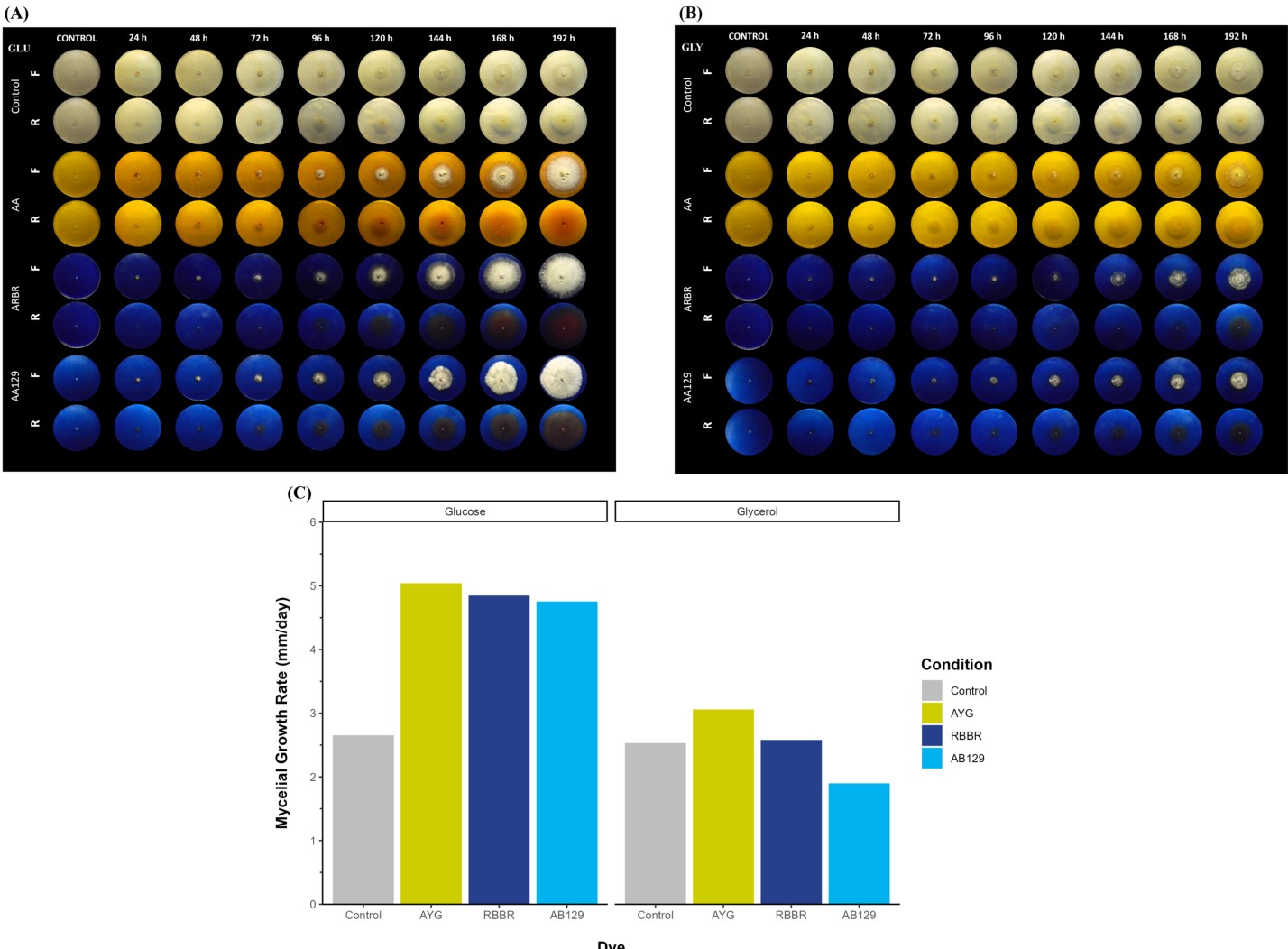

**Figure 1 Radial growth and decolorization of different azo and anthraquinone dyes by *P. ostreatus*.** The fungus was incubated on agar plates supplemented with glucose (A) or glycerol (B) as carbon source as well as complemented with either AYG, RBBR or AB129 dyes at 25 °C. Growth of the fungal colony and dye changes were monitored every 24 h for 8 days. Reverse (R) and front side (F). (C) Mycelial growth rate. Linear growth rates (mm/d) of the fungal colony, Kruskal-Wallis chi-squared = 4.083, $p$ = 0.0433, Averaged $R^2$ = 0.92.

# RESULTS

## Effect of carbon source on *P. ostreatus* growth and dye decolorization in plate assays

Figure 1A shows the growth and decolorization capacity of *P. ostreatus* on glucose and glycerol agar plates supplemented with 500 ppm of AYG, RBBR, or AB129 dyes. Although *P. ostreatus* grew on both glucose and glycerol media, a higher growth rate is observed on glucose, either alone or supplemented with the dyes. A statistically significant overall reduction in mycelial growth rates ($kr$) was observed on the plates containing either glucose or glycerol as the primary carbon source (Kruskal-Wallis chi-squared = 4.1325, df = 1, $p$-value = 0.04207). Moreover, adding dyes increased the growth rates on both

carbon sources, except for applying AB129 on glycerol, wherein the *kr* level decreased from 2.53 to 1.89 (Fig. 1B and Table S2).

This differential growth pattern led to the development of a larger fungal colony over the 192 h of incubation. The oxidation or transformation of all the dyes tested occurred simultaneously, along with the appearance of the mycelium, and varied by carbon source and type of dye, with the RBRR and AB129 dyes the most susceptible to decolorization and the AYG dye the least susceptible. Furthermore, a range of changes in the color of all the dyes was observed during the experiment. In the glucose media, the AYG dye transitioned from yellow to reddish hues over time, whereas such color alterations were not evident in the glycerol-supplemented medium. The changes observed in the RBBR and AB129 dyes were more evident than those observed for the AYG dye on both carbon sources. While AYG is a mono-azo dye with a complex structure and low redox potential, RBBR and AB129 are anthraquinone dyes. These latter dyes have a high redox potential and a less complex structure, which makes them more susceptible to decolorization. These findings suggest a significant influence of the carbon source and chemical nature of the dyes on both the growth kinetics of *P. ostreatus* and dye oxidation, thus highlighting the role of the carbon substrate in shaping fungal metabolism.

## Effect of glycerol on *P. ostreatus* growth, dye peroxidase activity and Acetyl Yellow G dye decolorization in submerged fermentation
### Characterization of growth kinetics and AYG decolorization

Submerged cultures were conducted to analyze the changes observed in the plate assays quantitatively. Figure 2 compares *P. ostreatus* growth in submerged fermentation using either glucose or glycerol as the single carbon source or as one supplemented with AYG dye. The variation in the maximal biomass (Xmax) obtained was higher for both the glucose and glucose/AYG cultures (7.09 and 7.20 g/L, respectively) than the glycerol or glycerol/AYG cultures evaluated (5.31 and 5.62 g/L, respectively). The same differences were observed for growth rate ($\mu$), with $\mu$ values of 0.033 and 0.047 h$^{-1}$ obtained for the glucose and glucose/AYG-media, values which are higher than the 0.027 and 0.023 h$^{-1}$ observed for the glycerol and glycerol/AYG media, respectively. Interestingly, adding the AYG dye did not significantly affect biomass production (Xmax), as no substantial differences were observed when the cultures with and without the presence of dye were compared.

The highest carbon consumption rate ($q_s$) was found in the presence of the AYG dye when glucose was used as the carbon source (Fig. 3). In effect, glucose depletion was observed at 240 h of culture, compared to 336 h for the fermentation conducted without dye, an expected finding considering the higher growth rate already reported. However, although the specific growth rates were similar in the fermentations with glycerol as carbon source, glycerol depletion was delayed when the dye was added (400 h), compared to 312 h for the fermentation conducted without the AYG dye. These findings suggest that with glucose, *P. ostreatus* can metabolize the carbon source more efficiently in the presence of the dye, leading to accelerated carbon source depletion. This effect was not observed in the

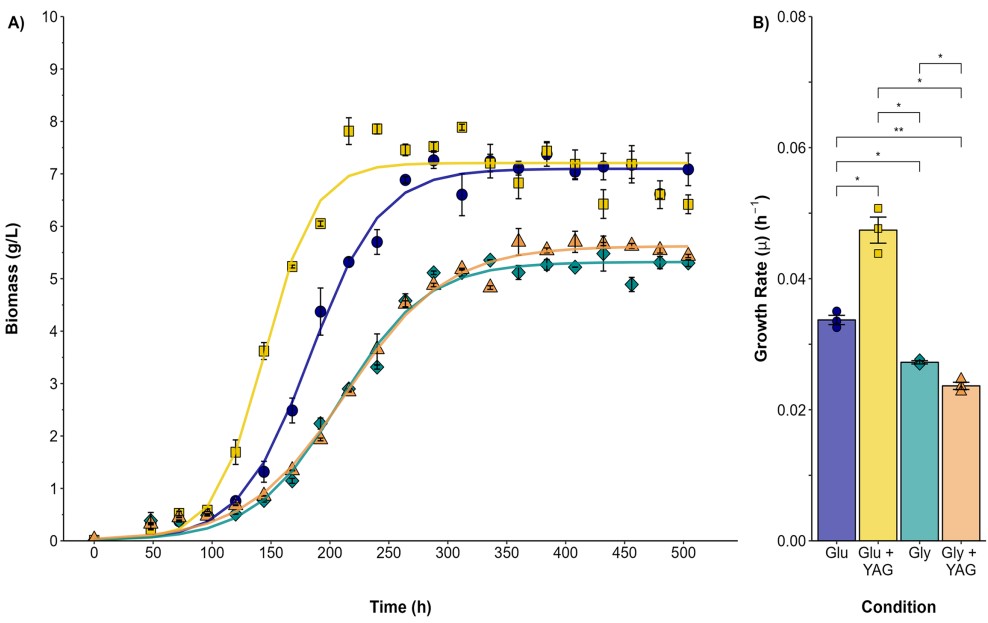

**Figure 2 Growth kinetics and specific growth rate values (μ) for *P. ostreatus* fermentations.** (A) Growth curve for glucose (blue circles), glucose supplemented with 500 ppm of AYG (yellow squares), glycerol (darkcyan diamonds), and glycerol supplemented with 500 ppm of AYG (orange triangles). Each point represents the mean of three replicates. The continuous line represents the best fit of the measured data to the logistic model. (B) Specific growth rate ($h^{-1}$) in each fermentation. Each bar represents the mean of three replicates. Statistical significance was calculated with *t-test* (*$p < 0.05$, **$p < 0.01$). The error bars in both panels represent the standard error.

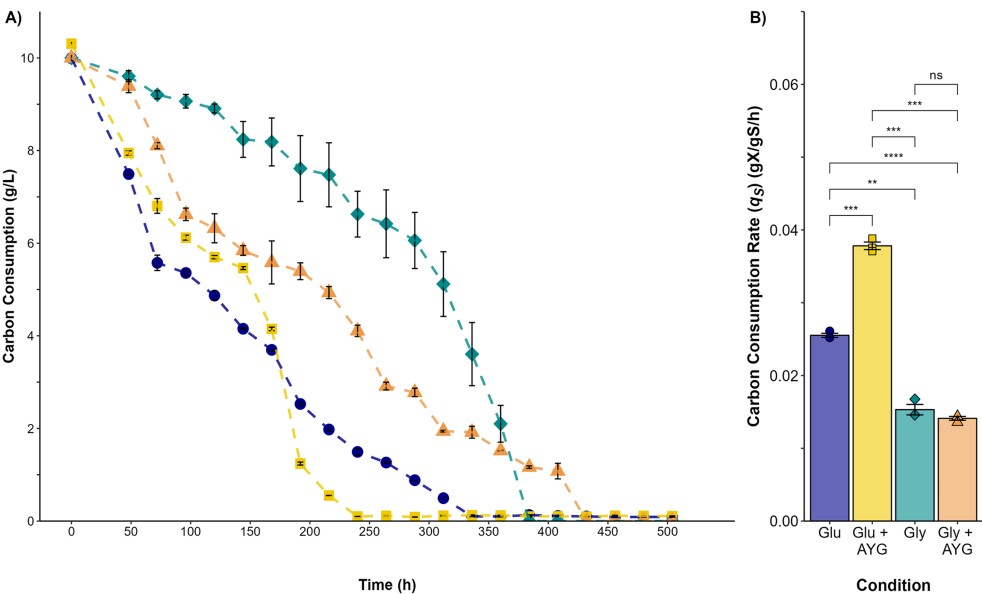

**Figure 3 Carbon source consumption.** (A) Consumption curves. During the fermentation with glucose (blue circle), glucose supplemented with 500 ppm of YAG (yellow squares), glycerol (green diamonds), and glycerol supplemented with 500 ppm of YAG (orange triangles). Each point represents the mean of three replicates. (B) Carbon consumption rate for each fermentation. Each bar represents the mean of three replicates. Statistical significance was calculated with *t-test* (ns = $p > 0.05$; **$p < 0.01$; ***$p < 0.001$; ****$p < 0.0001$). The error bars in both panels represent the standard error.

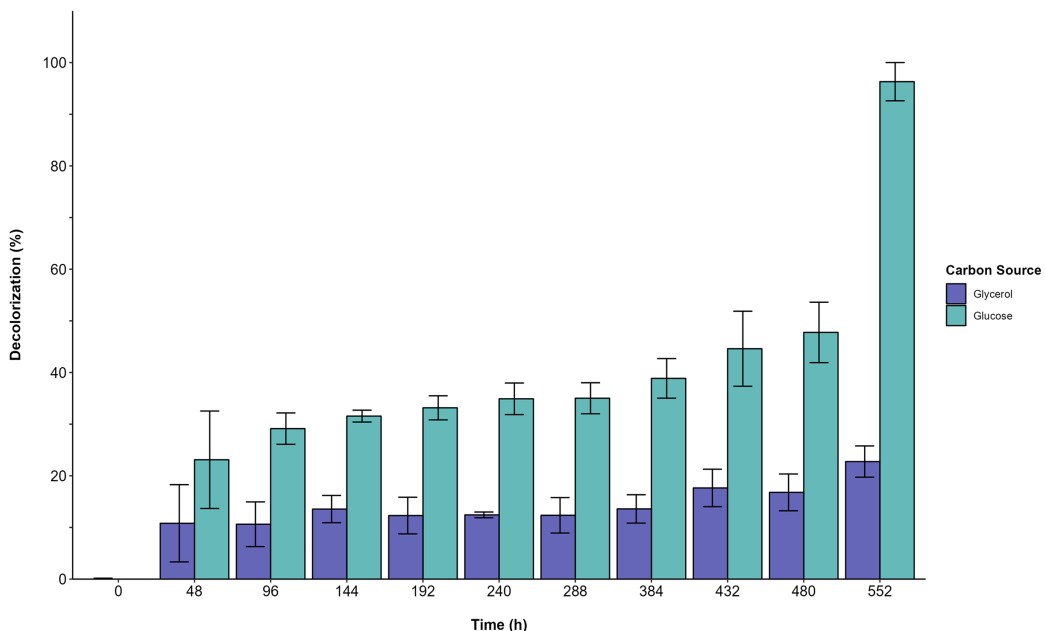

**Figure 4 Decolorization percentage.** Effect of carbon source on AYG dye decolorization during the time course of fermentation.

fermentations with glycerol, possible due to a more complex glycerol uptake and the catabolic pathways.

Furthermore, the decolorization percentage observed during glucose fermentations increased gradually throughout the fermentation, reaching percentages of 100% after 552 h. This finding contridicts the glycerol fermentation, wherein 10.8% decolorization was observed after only 48 h and a maximum of 22% decolorization was found after 552 h (Fig. 4). The UV/Vis changes in the absorption spectra found for the AYG dye coincided with the decolorization percentages observed for each fermentation (Fig. 5). The carbon source had a markedly discernible impact on the rate of dye decolorization.

### Effect of glycerol and AYG dye on DyP production

The effect of glycerol and AYG dye on dye peroxidase production by *P. ostreatus* is shown in Fig. 6. The highest titers of dye peroxidase activity (4,043 and 4,903 UI/L) were observed when glycerol and glycerol with AYG were used as a carbon source, reaching maximum levels at 408 and 360 h, respectively. On the other hand, the lowest activity levels were obtained for the glucose and glucose with AYG cultures (1,551 and 2,882 UI/L, at 312 and 288 h, respectively). Notably, irrespective of the carbon source, the addition of AYG dye consistently induced the production of DyP, as can be concluded from the higher activity levels observed early in the fermentation process. However, the highest DyP activity levels were mainly observed during the stationary growth phase (288 to 408 h). This finding does not concur with the decolorization percentages observed from 48 to 248 h, which may indicate the participation of other oxidases, such as laccase, manganese peroxidase, and versatile peroxidase, that are also produced by *P. ostreatus* and have oxidative potential for dyes.

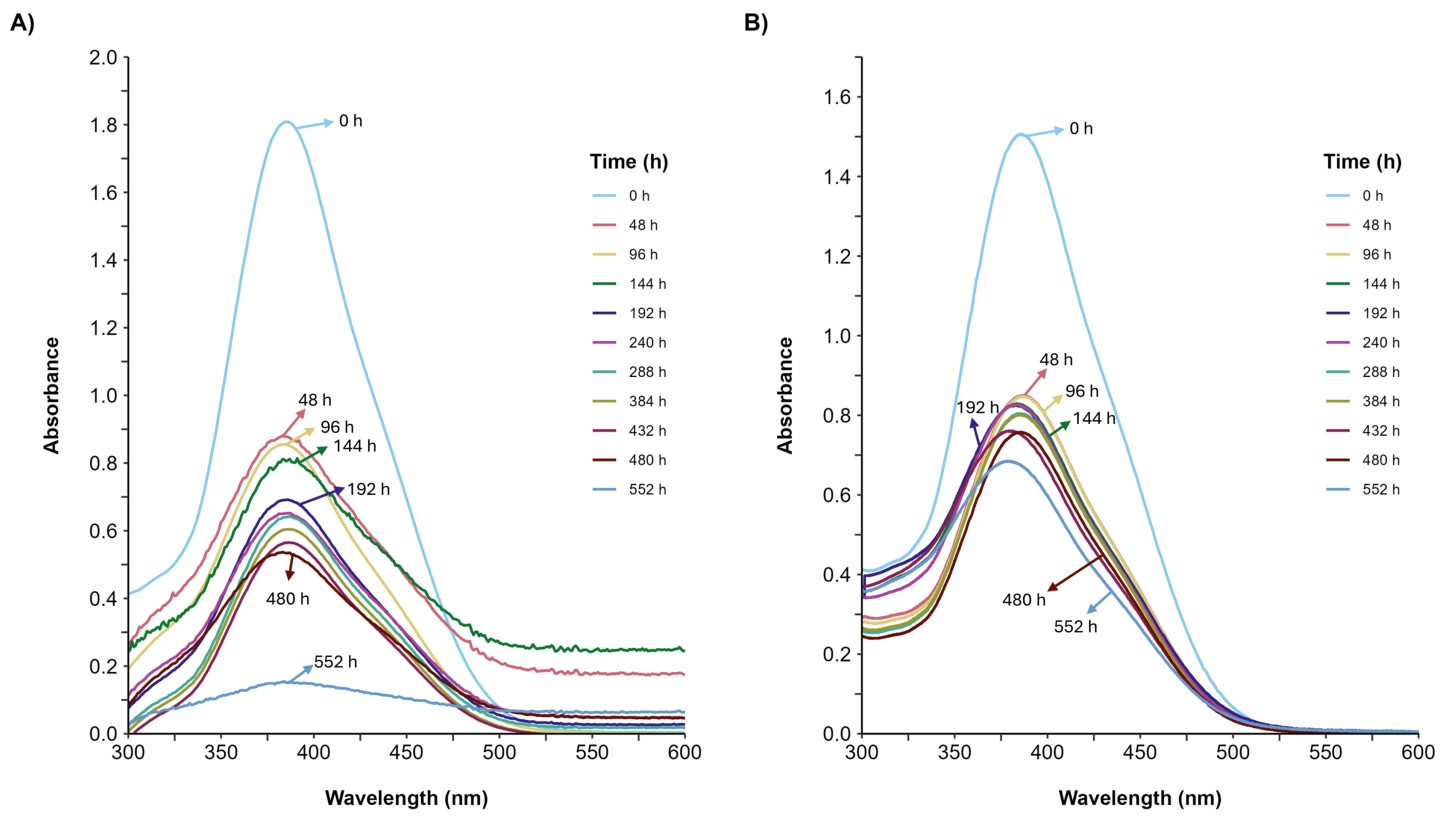

**Figure 5  UV/Vis spectral changes of AYG dye.** The UV/Vis spectra variation for the AYG was monitored during the *P. ostreatus* fermentation course, conducted at 25 °C for 552 h. (A) Glucose, (B) Glycerol. Maximum absorbance for AYG at 390 nm.

### Pleos-dyp genes expression and differential regulation in response to glycerol and AYG dye

The transcriptional response of *Pleos-dyp* genes to the use of glycerol as a carbon source and the addition of a synthetic dye (AYG) was also evaluated. Figures 7 and 8 show the influence of the carbon source and the AYG dye on the expression patterns of *dyp* genes, revealing a dynamic up-/down-regulation pattern throughout the fermentation process for the three *dyp* genes evaluated (*Pleos-dyp1*, *Pleos-dyp2*, and *Pleos-dyp4*).

The exploratory analysis of the global expression is shown in Fig. 7. The aim was to obtain an expression map of *P. ostreatus dyp* genes under the study conditions. The highest induction levels found for the control media were for the *Pleos-dyp1* and *Pleos-dyp4* genes, with 11.12 and 8.28-fold increases observed in the relative expression levels ($\log_2$) after 360 and 168 h, respectively. Additionally, gene expression profiles indicated that glycerol induced *Pleos-dyp1* and *Pleos-dyp2* genes, with a 11.61- and 4.28-fold increase observed after 144 h, respectively. On the other hand, the addition of AYG resulted in a respective 6.69 and 3.59-fold increase in the induction of glucose after 504 h for *Pleos-dyp4* and *Pleos-dyp1*, respectively, and a 12.86 and 4.02-fold increase in induction levels for glycerol, after 360 and 504 h of culture for *Pleos-dyp1* and *Pleos-dyp2*, respectively. Interestingly, in the fermentations with glycerol, the expression of *Pleos-dyp4* was not detected.

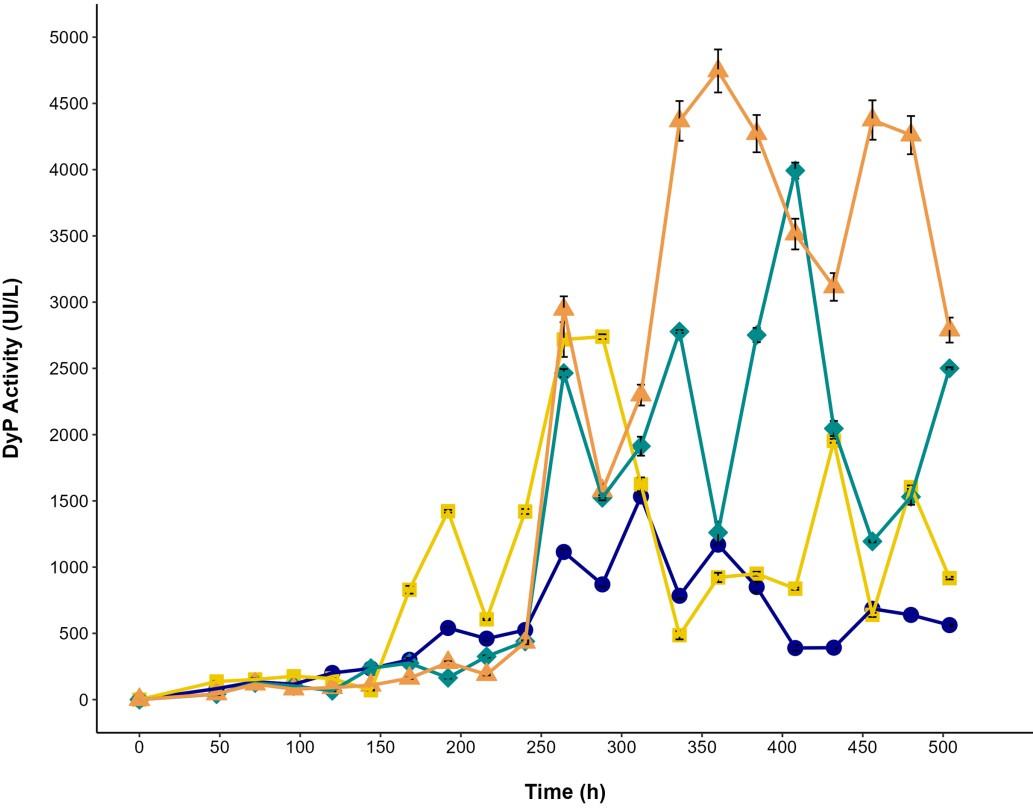

**Figure 6 DyP activity in the cultures of *P. ostreatus*.** Enzymatic activity during the time-course of *P. ostreatus* fermentation with glucose (blue circle), glucose supplemented with 500 ppm of AYG (yellow squares), glycerol (green diamonds), and glycerol supplemented with 500 ppm of AYG (orange triangles). Each point represents the mean of three replicates. Error bars indicate the standard error.

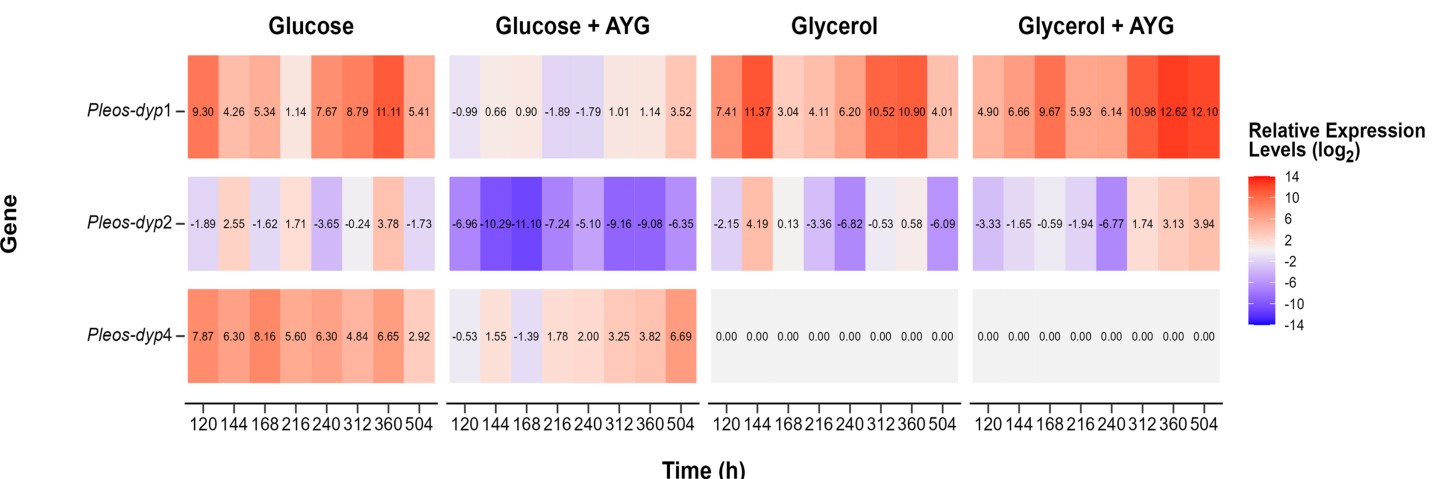

**Figure 7 Heat map of global gene expression.** Exploratory analysis of gene expression of the three *Pleos-dyp* genes during the time-course fermentation with Glucose, Glycerol, and Glucose or Glycerol Supplemented with 500 ppm of AYG. The reference gene (*pep*) was used for data normalization.

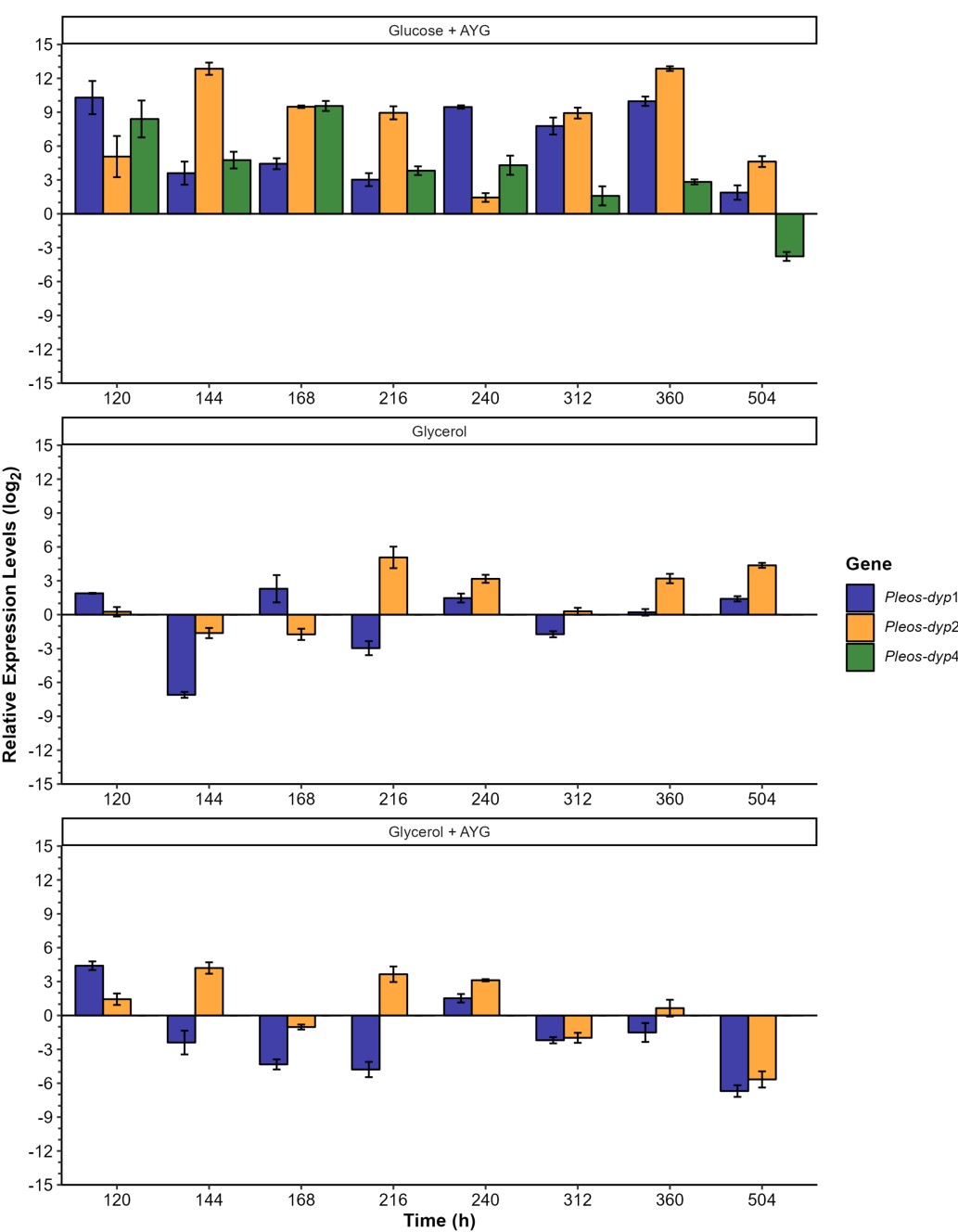

**Figure 8 Effect of carbon source and AYG dye on the relative expression levels of the three *Pleos-dyp* genes.** The expression patterns of the genes *Pleos-dyp1*, *Pleos-dyp2* and *Pleos-dyp4* in response to the carbon source (glucose or glycerol) and the addition of AYG dye during the time-course of a submerged *Pleurotus ostreatus* fermentation. The increased or decreased expression values were referred to those obtained under the control condition (glucose without dye). Error bars represent the standard deviations of the means of three independent amplifications.

The effects of the different carbon sources, AYG dye and fermentation times on the dye peroxidase expression were also analyzed by referring to the transcriptional levels observed to those obtained under the control media (glucose) (Fig. 8). Adding AYG led to

up-regulation (an increase of the enzymatic activity) of the three *dyp* genes throughout the fermentation with glucose. The highest induction observed was found for *Pleos-dyp2* and *Pleos-dyp1* genes, equivalent to 12.85 and 10.29 fold-increase at 144 and 120 h, respectively. This effect was also observed in fermentation with glycerol, but to a lesser extent. After 120 and 144 h of culture, 4.41 and 4.20 fold increases were observed in the induction levels presented by *Pleos-dyp1* and *Pleos-dyp2* respectively. Concerning the glycerol effect, in those fermentations the expression levels showed up- and down-regulation, with similar expression patterns observed for *Pleos-dyp1* and *Pleos-dyp2* observed.

These findings underscore the intricate dynamics of gene expression in response to different carbon sources and the presence of AYG dye, shedding light on the regulatory mechanisms governing dye peroxidase production in *P. ostreatus.*

## DISCUSSION

Within the *P. ostreatus* genome, four dye-decolorizing peroxidase genes have been identified (*Ruiz-Dueñas et al., 2011*). According to *Fernández-Fueyo et al. (2015)*, only two out of four *Pleos-DyPs* were found to be phylogenetically divergent (*Pleos-DyP1* and *Pleos-DyP4*). However, Pleos-DyPs isoenzymes differ phylogenetically and in terms of their dye redox potential. Limited reports on factors regulating DyP synthesis are available in the literature. For instance, the effect of a less-metabolizable carbon source, such as glycerol, on the differential expression of DyP-encoding genes and the corresponding activity produced is relevant to understanding the mechanisms involved in DyP production. The high number of genes in the *P. ostreatus* genome encoding DyP and other oxidase isoenzymes indicates diversity in their properties and differential regulation, as discussed below.

### Effect of carbon source on growth

Glycerol can be used as a carbon and energy source for several groups of fungi; however, its utilization efficiency varies among fungi, and compared to glucose, the glycerol metabolism is often less efficient (*Klein et al., 2017*; *Liu, Jensen & Workman, 2012*; *Urek & Kasikara Pazarlioglu, 2007*). The present research first evaluated the effect of glycerol as a carbon source on *P. ostreatus* growth, in both plate assays and submerged fermentation. Glycerol showed lower efficiency as a carbon source than glucose, significantly affecting both growth rate and biomass production. The findings of the present study confirm that previous research demonstrated that *P. ostreatus* growth is affected by using a complex carbon source. such as glycerol, resulting in reduced biomass production and altered growth rates (*Mikiashvili et al., 2006*; *Sarris et al., 2020*; *Tinoco et al., 2011*). Glycerol may undergo a more complex catabolic process than glycolysis, the catabolic process undergone by glucose. The phosphorylative glycerol catabolic pathway is widespread among fungi and requires the formation of L-glycerol 3-phosphate and dihydroxyacetone phosphate before glycolysis or gluconeogenesis can proceed (*Klein et al., 2017*).

Additionally, it has been suggested that the deficient regulation of the enzymes involved in the primary steps of glycerol catabolism negatively influences the growth of several types
of fungi when glycerol is used as the sole carbon source (*Papanikolaou et al., 2017*; *Sarris et al., 2020*). The results obtained by the present study show that, particularly with glucose used as the carbon source, the addition of the dyes increased both growth and carbon-source-consumption rates. This finding concurs with other reports in the literature, which found that the addition of exogenous aromatic compounds to white-rot basidiomycetes has been shown to induce the production of glycolytic enzymes within other enzymes involved in sugar metabolism, thus resulting in the activation of glucose consumption (*Shimizu et al., 2005*).

## Effect of glycerol and AYG dye on the production of DyP activity and dye decolorization

The present study found that using glycerol instead of glucose resulted in a three-fold increase in DyP activity. Similar observations were reported by *Roch et al. (1989)*, who found that *Phanerochaete chrysosporium* could grow under carbon limitation with glycerol used as a carbon source, thus both affecting *P. chrysospoorium* growth and increasing lignin peroxidase activity. Many studies on the effects of medium composition on lignin-modifying enzyme (LME) production have focused on optimizing laccase activity or global LME induction, rather than specifically examining DyPs. *Tinoco et al. (2011)* optimized a culture medium for laccase production by *P. ostreatus*, using copper and lignin as inducers. In contrast to the findings presented here, they did not observe the significant influence of glucose or glycerol on laccase production. Various reports have indicated that the production of LMEs in basidiomycetes is dependent on the carbon and nitrogen sources used, as well as the presence of aromatic compounds in the culture medium (*Elisashvili, Kachlishvili & Asatiani, 2018*; *Stajić et al., 2006*; *Thiribhuvanamala et al., 2017*). For instance, *Elisashvili et al. (2002)* demonstrated the effect of different carbon sources and aromatic compounds on the lignocellulolytic enzyme activity of different edible and medicinal basidiomycetes, concluding that it is possible not only to increase lignocellulolytic activity substantially, but also to trigger their preferential synthesis by supplementing the culture medium with nutritional compounds. According to *Jiao et al. (2018)*, adding small aromatic molecules to the culture medium could increase the laccase yield, thus activating laccase gene transcription by binding onto target gene's xenobiotic response element (XRE). The present study found that the production of DyP activity was primarily observed during the stationary phase (268–552 h) of the *P. ostreatus* culture. At the same time, adding AYG dye (an azoic dye whose chemical composition includes aromatic rings, azoic linkages, and amino and sulphonic groups) increased maximum activity levels. Ligninolytic enzymes are generally produced as secondary metabolites (*Elisashvili, Asatiani & Kachlishvili, 2020*; *Thiribhuvanamala et al., 2017*). It is assumed that the metabolized substrate is essential for fungi not only for synthesizing lignin-degrading enzymes but also, for producing peroxide and the effectors of the ligninolytic system (*Shimizu et al., 2005*). According to *Buswell, Mollet & Odier (1984)*, growing *Phanerochaete chrysosporium* on glycerol leads to carbon limitation, which can affect the onset of the secondary metabolism, a condition reported to favor the carbon catabolite de-repression of both CAZ and ligninolytic enzymes (*Peng et al., 2021*; *Suzuki,*

*Igarashi & Samejima, 2008*). *Aro, Pakula & Penttilä (2005)* reported that the depletion of nutrient nitrogen, carbon, or sulfur generally triggers the expression of gene encoding ligninolytic enzymes. Furthermore, the addition of AYG increased the carbon source consumption rate. It is known that, in the presence of a primary carbon source, the co-metabolism of different pollutants, such as dyes, is widely observed in fungi and other microorganisms. The co-metabolism of the pollutants can be accomplished by enzymes that convert them into intermediates that the microorganism can then use. One of the mechanisms proposed as participating in this process involves both the primary carbon source and the co-substrate acting as co-inducers in the activation of different genes that produce the enzymes that act on the carbon source and the pollutant (*Ahlawat, Jaswal & Mishra, 2022*; *Shimizu et al., 2005*). In this sense, several reports indicate that the efficiency of dye decolorization can be favored by co-metabolism with different carbon sources, of which glucose, sucrose, fructose, and glycerol are among the most extensively studied (*Civzele, Stipniece-Jekimova & Mezule, 2023*; *Haider et al., 2019*; *Merino, Eibes & Hormaza, 2019*; *Rao et al., 2019*). The results obtained for the analysis conducted on carbon source consumption and DyP activity levels for both carbon sources analyzed in the present research suggest that the AYG dye may be co-metabolized, thus inducing the production of DyP and other oxidases not evaluated in the experiments conducted by the present study. The ability of *P. ostreatus* to metabolize a wide variety of toxic compounds is primarily attributed to their non-specific multi-enzyme oxidative system, which comprises manganese peroxidases (MnPs; EC1.11.1.13), versatile peroxidases (VPs (EC1.11.1.16), laccases (Lacs; EC1.10.3.2), and dye-decolorizing peroxidases (DyP;EC1.11.1.19) (*George et al., 2023*; *Kunjadia et al., 2016*; *Šlosarčíková et al., 2020*). Our previous research reported that, when glucose was the sole carbon source, the addition of Acetyl Yellow G (AYG), Remazol Brilliant Blue R (RBBR), or Acid Blue 129 (AB129) dyes increased DyP activity, ultimately achieving complete decolorization (*Cuamatzi-Flores et al., 2019*). When grown in liquid media, the transformation of RBBR dye by *P. ostreatus* seems to occur mainly *via* laccase oxidation. However, dye-decolorizing peroxidase and veratryl alcohol oxidase were also produced (*Palmieri, Cennamo & Sannia, 2005*). *Eichlerová, Homolka & Nerud (2006)*, evaluated the decolorization capacity of Orange G and Remazol Brilliant Blue R (RBBR) dye and the ligninolytic-enzyme production of eight different *Pleurotus* species. The main enzymes detected were Lac and MnP, whose production was strongly influenced by the cultivation media type and the dye's presence. *Ottoni et al. (2014)* reported that, in *Trametes versicolor*, glycerol is an important substrate for oxidative metabolism, promoting higher laccase production and, increasing the decolorization of Reactive Black 5.

The efficiency of AYG decolorization was considerably affected by the carbon source. The present study found 45% and 10% decolorization percentages after 48 h in glucose and glycerol, respectively. Furthermore, in glucose, complete decolorization was observed after 552 h. It has been reported that dye decolorization or degradation efficiency depends on different factors, such as the carbon source or chemical nature of the dyes. The degradation of dyes involving aromatic cleavage depends on the type, number, and position of functional groups, and other factors, such as electron distribution and charge density.

Consequently, color removal is less efficient with highly substituted dyes and presents higher molecular weights (*Lu et al., 2008*; *Tochhawng et al., 2019*). As observed in plate assays and submerged fermentation, a poorer decolorization of the mono-azo dye AYG was found than that observed for the anthraquinone dyes (ABBR and AB129), with both dyes presenting a higher redox potential and a less complex structure than AYG.

### *Pleos-dyp* gene expression and differential regulation in response to glycerol and AYG dye

Several studies conducted on Basidiomycota have demonstrated that CCR, in combination with different signaling pathways, plays a key role in the utilization of different carbon sources in this group of fungi (*Hu et al., 2020*; *Nakazawa et al., 2019*; *Toyokawa et al., 2016*; *Zhang et al., 2022*). The existence of an ortholog of Cre1, the main transcriptional regulator in the CCR pathway, has also been demonstrated and may participate in this regulatory process (*Alfaro et al., 2020*; *Pareek et al., 2022*; *Yoav et al., 2018*). Furthermore, it has been shown that Cre1 is regulated by cAMP-dependent protein kinase A (PKA) (*de Assis et al., 2020*; *Pareek et al., 2022*), with both Cre1 and PKA potentially involved in the induction of genes that encode lignin-modifying enzymes in *P. ostreatus* (*Toyokawa et al., 2016*).
The present study used glycerol as an alternative substrate to examine the transcriptional responses of *P. ostreatus dyp* genes and investigate the potential participation of carbon catabolite de-repression in their regulation. The analysis of *dyp* gene expression profiles conducted by the present research revealed significant variations influenced by the carbon source, growth phase, and the presence of AYG dye. These variations led to up- and down-regulation patterns in the three *Pleos-dyp* genes evaluated over the fermentation period. Interestingly, adding glycerol and AYG dye induced the early-stage expression of *Pleos-dyp1* (144 h). In contrast, the expression of *Pleos-dyp4* was not detected in the glycerol cultures. The potential XRE, Cre1, and Nit2 binding site motifs identified in the promoter of the three DyP encoding genes analyzed (Table S1) suggest that the transcription of the *dyp* gene, among others, can be regulated by xenobiotics and carbon and nitrogen sources. The frequency of these cis-acting elements varies among genes, from zero for XRE in the *dyp2* gene to one for Cre1 in the three genes and three for Nit2 in *dyp4*. The effect of chemical dyes on the DyP activity and gene expression profile of *P. ostreatus* has been previously reported. Adding dyes results in an induction effect on the enzyme activity and the expression profiles of *dyp* genes, with maximum induction levels detected for the *dyp4* gene at the end of the fermentation process (*Cuamatzi-Flores et al., 2019*). Further studies must be conducted to define the DyP regulation occurring *via* CCR and validate the functionality of the cis-regulatory elements identified in the *dyp* gene promoters.

## CONCLUSIONS

The carbon source impacted the growth, DyP production, and differential regulation of *dyp* genes in *P. ostreatus*. Glycerol is used as a carbon source, but it negatively affects growth rate and carbon consumption. However, DyP production was significantly higher than that observed with glucose as a carbon source. Adding the AYG dye increased DyP

production at different levels depending on the carbon source. The expression of the three *dyp* genes was differentially affected by the carbon source and the addition of the AYG dye. The *Pleos-dyp1* and *Pleos-dyp2* genes showed the highest transcription levels, while *Pleos-dyp4* expression was not detected in the glycerol sample. With the addition of the AYG dye, particularly in the glucose culture, the three *Pleos-dyp* genes were upregulated throughout the fermentation. Although glycerol increased DyP activity, the transcript levels observed did not correlate with DyP activity, particularly in the stationary phase of the fermentation.

### Funding

This work was supported by the Instituto Politécnico Nacional, (SIP No. 20231995) and Comisión de Operación y Fomento de Actividades Académicas (COFAA-IPN). The funders had no role in study design, data collection and analysis, decision to publish, or preparation of the manuscript.

### Grant Disclosures

The following grant information was disclosed by the authors:
Instituto Politécnico Nacional: 20231995.
Comisión de Operación y Fomento de Actividades Académicas (COFAA-IPN).

### Competing Interests

The authors declare that they have no competing interests.

### Author Contributions

- Jorge Cuamatzi-Flores performed the experiments, analyzed the data, prepared figures and/or tables, and approved the final draft.
- Soley Nava-Galicia performed the experiments, analyzed the data, prepared figures and/or tables, and approved the final draft.
- Edgardo Ulises Esquivel-Naranjo conceived and designed the experiments, analyzed the data, authored or reviewed drafts of the article, and approved the final draft.
- Agustin Lopez Munguia conceived and designed the experiments, analyzed the data, authored or reviewed drafts of the article, and approved the final draft.
- Analilia Arroyo-Becerra conceived and designed the experiments, analyzed the data, authored or reviewed drafts of the article, and approved the final draft.
- Miguel Angel Villalobos-López conceived and designed the experiments, analyzed the data, authored or reviewed drafts of the article, and approved the final draft.
- Martha Bibbins-Martínez conceived and designed the experiments, analyzed the data, prepared figures and/or tables, authored or reviewed drafts of the article, and approved the final draft.

### Data Availability

The data is available at Zenodo: Cuamatzi-Flores, J. (2024). Regulation of Dye-decolorizing Peroxidases Gene Expression in *Pleurotus ostreatus* Grown on Glycerol as the Carbon Source [Data set]. Zenodo. https://doi.org/10.5281/zenodo.10631490.

## Supplemental Information

Supplemental information for this article can be found online at http://dx.doi.org/10.7717/peerj.17467#supplemental-information.

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
