# Peer review of "Regulation of dye-decolorizing peroxidase gene expression in Pleurotus ostreatus grown on glycerol as the carbon source"

_PeerJ, doi:10.7717/peerj.17467_

## Round 0.1 · original submission · Major Revisions

You have received a request to provide extensive comments and suggestions for a major revision of a manuscript. One important correction to be made is regarding the name for dye-decolorizing peroxidases, which should be "DyP" instead of "DyeP" as mentioned in the title. Please review and make necessary corrections to the manuscript and the title. Additionally, it is required to include a detailed rebuttal letter for each reviewer.

Reviewer 1 ·

Basic reporting

In this research, the authors analyzed the effect of glucose and glycerol with or without the dye Acetyl Yellow G on growth Pleurotus ostreatus growth, decolorization, total DyePs activity and the expression of three -DyeP genes.
It is indeed interesting and useful to study glycerol as carbon source.

Experimental design

The methods are described in a clear way.

It could be very interesting to compare the Acetyl Yellow G results to dyes from other families.

Validity of the findings

The finding are valid

The discussion is interesting but very long and deals with issues not studied in the research – the role of laccase and potential regulation by CCR can be shortened

Additional comments

Fig 1 is difficult to understand. I could not see the decolorization in the glycerol plates. It is better to give the numerical data (dimeter or area of the colony and the decolorization zone)

Fig 2. I did not see explanation for the faster growth in the glucose treatment in the presence of the dye – is it a carbon or nitrogen source?
It is in some contradiction with the C source consumption described in fig 3.
It is indeed very good to give a time course data. I could not see a correlation between the decolorization and total enzyme activity. Are there additional enzymes involved?

·

Basic reporting

The article is very interesting, but needs to make some aspects clearer. For example, a modified methodology from a reference needs to be presented highlighting its modifications.

Another important point is references. It has phrases with countless references and some very old. It is not necessary to put so many references in one sentence. The article presents around 70 references.

Experimental design

Methods described with sufficient detail & information to replicate. Not for example:

L 138 The total RNA was isolated from frozen mycelia harvested at different time-points of the
139 fermentation, using the NTES extraction protocol.
What is the reference? what is it NTES extraction protocol?

L 148 The reaction mixture, the amplification program, the melting curve, and the selection of the reference genes were adapted from Garrido-Baz·n et al., (2016).
What was adapted? What were the changes?

Table 1 '...the selected reference gene" please make it clear which is the reference gene in the table 1.

Supplementary Table 1. Analysis of regulatory cis-elements in promoter regions of Pleos-DyeP genes. Occurrence of putative cis-acting regulatory elements (TATA-box, CAAT-box, carbon catabolite repressor binding elements (CreA), metal responsive element (MRE), xenobiotic responsive element (XRE), nitrogen binding site (NIT2).
This analysis is not described in the methodology.

Validity of the findings

All underlying data have been provided; they are robust, statistically sound, & controlled.


Conclusions
I believe that the conclusion should be based on the objectives of the article. The authors expand the conclusions based on non-detailed analyses.
For example:

The analysis of the promoters of DyeP encoding genes has revealed the presence of several putative cis-regulatory elements, including the CCR Cre1-binding site.

Additional comments

It is not clear why there are no glucose + AYG treatments in figure 6?

review lines 189 to 192. I believe I have an error in interpreting figure 3.
review line 203. I believe I have an error in interpreting figure 5.

Reviewer 3 ·

Basic reporting

The manuscript entitled Regulation of dye peroxidase gene expression in Pleurotus ostreatus grown on glycerol as the carbon Source by Jorge Cuamatzi-Flores, Soley Nava-Galicia, Edgardo Esquivel-Naranjo, Agustín López-Munguía, Analilia Arroyo-Becerra, Miguel Angel Villalobos-López and Martha Bibbins-Martínez describe the effects of the carbon source, specifically glycerol, over the expression and putative regulation of several dye peroxydases from Pleurotus ostreatus using as a main evidence of their conclusions RT-qPCR experiments. In general, the paper is well written, and the use of references is sufficient to follow the manuscript and, in general terms, to follow their arguments and speculations to draw their conclusions.

Experimental design

The experimental design is professional, however, the results to support the conclusions are preliminary results and need to be supported with more data to improve the clarity of their results and reduce speculation. In the manuscripts authors reference several times their paper published in 2019 in PlosOne, however, in this publication, the addition to AYG increases the expression, enzymatic activity, and decolorization capacity of Dye 1-4, but expression profiles of dye 1 and 2 in the presence of glucose are very similar, in contraposition to the results present in their new contribution. Even the authors observed an improved expression in the presence of glycerol as a carbon source, with a concomitant improved expression of dye 1-4, these results do not explain with enough clarity the decolorization results. In the discussion authors briefly mention the glycerol effect over decolorization results, which needs to be expanded in a new version of the manuscript. In general, the results do not fit with the expectative draw in the introduction, affecting the discussion and increasing the speculations in the text, these generate doubts over the quantity and quality of the results in the whole manuscript. Additionally, the catabolic repression hypothesis presented in the paper seems very preliminary and strongly speculative.

Validity of the findings

The paper in the current version is thus very preliminary, and to be considered for publication needs to reduce speculations by adding extra data like expression profiles using SDS-Phage or substantially improving and expanding the RT-qPCR results, these also imply a restructuration of figures.
A list of minor issues are listed below:

1. Line 68. The letter “S” needs to be added to the word play.
2. Line 140. Please add “nm” after 260/280

---

## Round 0.2 · Minor Revisions

Please attend to the recommendations of the reviewer and submit an improved version.

·

Basic reporting

no comment

Experimental design

no comment

Validity of the findings

no comment

Additional comments

The article was very well reviewed by the authors. Here are just two suggestions below:

1) L109 Petri dishes containing agar 15g/L, glucose, or glycerol as the carbon source and 500 ppm of
Sugestion: L109 Petri dishes containing agar 15g/L, glucose (G or Glu), or glycerol (Gly) as the carbon source and 500 ppm

2) corrections to figure 1 caption

(A)The fungus was incubated on agar plates supplemented with glucose or glycerol as carbon source as well as complemented with either AYG, RBBR or AB129 dyes at 25 °C. Growth of the fungal colony and dye changes were monitored every 24 h for 8 days. Reverse (R) and front side (F). (B) Mycelial growth rate. Linear growth rates (mm/d) of the fungal colony, Kruskal-Wallis chi-squared = 4.083, p = 0.0433, Averaged R2 =0.92.

Sugestion: The fungus was incubated on agar plates supplemented with glucose (A) or glycerol (B) as carbon source as well as complemented with either AYG, RBBR or AB129 dyes at 25 °C. Growth of the fungal colony and dye changes were monitored every 24 h for 8 days. Reverse (R) and front side (F). (C) Mycelial growth rate. Linear growth rates (mm/d) of the fungal colony, Kruskal-Wallis chi-squared = 4.083, p = 0.0433, Averaged R2 =0.92.

---

## Round 0.3 · Minor Revisions

A file containing suggestions for style changes is included. Please consider these changes before submitting the revised version. It may be helpful to consult a professional editing service for further improvements.

---

## Round 0.4 · accepted · Accept

Thanks for addressing the revisions requested. Now, your manuscript is accepted in PeerJ.